# Epidemiological and Genetic Characterization of Coxsackievirus A6-Associated Hand, Foot, and Mouth Disease in Gwangju, South Korea, in 2022

**DOI:** 10.3390/v16030476

**Published:** 2024-03-20

**Authors:** Ji-Eun Lee, Min-Ji Kim, Mi-Hyeon Lim, Sue-Ji Han, Jin-Yeong Kim, Soo-Hoo Kim, Yi-Duen Ha, Gyung-Li Gang, Yoon-Seok Chung, Jung-Mi Seo

**Affiliations:** 1Health and Environment Research Institute of Gwangju, Gwangju 61954, Republic of Korea; vetmj@korea.kr (M.-J.K.); limmh78@korea.kr (M.-H.L.); hsjhj94@korea.kr (S.-J.H.); jiny0805@korea.kr (J.-Y.K.); suh00519@korea.kr (S.-H.K.); cuteyideng@korea.kr (Y.-D.H.); ggl21@korea.kr (G.-L.G.); sm6698@korea.kr (J.-M.S.); 2Division of High-Risk Pathogen, Bureau of Infectious Diseases Diagnosis Control, Korea Disease Control and Prevention Agency (KDCA), Cheongju 28159, Republic of Korea

**Keywords:** HFMD, coxsackievirus A6, molecular epidemiology, phylogenetic tree

## Abstract

Coxsackievirus A6 (CV-A6) has emerged as the predominant causative agent of hand, foot, and mouth disease (HFMD) in young children. Since the declaration of coronavirus disease 2019 (COVID-19) as a global pandemic, the incidence of infectious diseases, including HFMD, has decreased markedly. When social mitigation was relaxed during the COVID-19 pandemic in 2022, the re-emergence of HFMD was observed in Gwangju, South Korea, and seasonal characteristics of the disease appeared to have changed. To investigate the molecular characteristics of enterovirus (EV) associated with HFMD during 2022, 277 specimens were collected. Children aged younger than 5 years accounted for the majority of affected individuals. EV detection and genotyping were performed using real-time RT-PCR and nested RT-PCR followed by sequence analysis. The EV detection rate was found to be 82.3%, and the main genotype identified was CV-A6. Sixteen CV-A6 samples were selected for whole genome sequencing. According to phylogenetic analysis, all CV-A6 strains from this study belonged to the sub-genotype D3 clade based on VP1 sequences. Analysis of 3D polymerase phylogeny showed that only the recombinant RF-A group was identified. In conclusion, circulating EV types should be continuously monitored to understand pathogen emergence and evolution during the post-pandemic era.

## 1. Introduction

Hand, foot, and mouth disease (HFMD) is an acute viral infectious disease that mostly affects humans in early childhood (children younger than 5 years of age). The typical symptoms of HFMD include fever, rashes, or blisters on the hands and feet, and mouth sores. HFMD is primarily caused by a group of human enteroviruses (HEVs), in particular HEV-A, including coxsackievirus A16 (CV-A16) and enterovirus A71 (EV-A71) [1]. Although CV-A16 and EV-A71 have historically been considered the leading pathogens causing HFMD across the Asia-Pacific region [2,3,4], CV-A6-related HFMD outbreaks have occurred frequently worldwide in recent years. Since the initial outbreak in Finland in 2008 [5], numerous CV-A6-related HFMD outbreaks have occurred in European countries [6,7,8], America [9], Singapore [10], Hong Kong [11], Japan [12], Thailand [13,14], and China [15,16,17]. Therefore, rapidly spreading HFMD cases caused by CV-A6 infections have markedly increased the public health burden. CV-A6 has gradually played a key role in the outbreak of HFMD, thereby replacing CV-A16 and EV-A71 as the major causative agent of HFMD; thus, CV-A6-associated HFMD has become an important part of enterovirus surveillance.

Since the announcement of coronavirus disease 2019 (COVID-19) as a global pandemic in early 2020, the incidence of viral infectious diseases, including HFMD, had decreased considerably. However, when social mitigation was relaxed in 2022, HFMD epidemics caused by CV-A6 infections were first detected in Gwangju during the COVID-19 pandemic. This study aimed to elucidate the molecular and epidemiological characteristics of CV-A6-associated HFMD cases diagnosed in surveillance hospitals across Gwangju in 2022. Toward this end, we analyzed CV-A6-positive samples using next-generation sequencing and described the genetic characteristics of CV-A6 in Gwangju.

## 2. Materials and Methods

### 2.1. Specimen Collection

Annual reports on HFMD are accessible to the public on the website of the Korea Disease Control and Prevention Agency (KDCA). Weekly national surveillance data on HFMD from 2019 to 2022 were obtained from the KDCA Infectious Disease Web Portal “https://www.kdca.go.kr/npt (accessed on 14 September 2023)”. This surveillance was conducted via a network of pediatric hospitals that reported all cases of EV infection, including HFMD. The Korea Enterovirus Surveillance System has been implemented by regional health and environment research institutes and medical institutions. We participated in a national surveillance program to monitor EV circulation in Gwangju, South Korea. All clinical specimens (including feces, throat swabs, and nasopharyngeal swabs) from patients presenting with HFMD were collected by the hospitals in the Gwangju area and sent weekly to the Institute of Health and Environment Research for detecting HEV.

### 2.2. Detection and Identification of CV-A6

HEV RNA was extracted from the specimens using the QIAamp viral RNA mini kit (Qiagen, Hilden, Germany) according to the manufacturer’s instructions. To directly detect HEVs, real-time reverse transcription-PCR (RT-PCR) was performed firstly by using the human EV/EV71 multiplex real-time PCR kit (Kogenebiotech, Seoul, Republic of Korea). Real-time RT-PCR amplifications were run on the 7500 Fast Real-Time PCR system (Applied Biosystems, Foster City, CA, USA), and PCR was conducted according to the manufacturer’s instructions. Samples positive for HEV were subjected to nested RT-PCR for identification of the HEV genotypes. VP1 gene sequences were amplified using the VP1 1st RT-PCR kit and VP1 2nd PCR kit (iNtRON Biotechnology, Gyeonggi, Republic of Korea). Primers used in first PCR were as follows: forward primers 5′-GCR ATG TTR GGR ACW CAT GT-3′; 5′-GCS ATG TTR GGM ACR CAY GT-3′ and reverse primer 5′-GGR TTB GWK GAN GTY TGC CA-3′. The primer combination used in the semi-nested PCR consists of two forward primers and two reverse primers. Primers used in secondary PCR were as follows: forward primers 5′-CCH GCD CTH ACC GCW GTG GAR ACD GG-3′; 5′-CCM ATM CTH CAA GCH GCH GAG GAG AYY GG-3′ and reverse primers 5′-GGR SCN CCD GGW GGY ACA WAC AT-3′; 5′-GGH GCV CCY GGY GGY ACR TAC AT-3′.

All PCR products were subsequently sequenced in both forward and reverse directions. Obtained sequences were assembled using BioEdit software (version 7.7.1) and we then compared sequence homology with other enterovirus sequences in GenBank

### 2.3. Complete Genome Sequencing

Sixteen HEV samples underwent whole genome sequencing. A customized Ion AmpliSeq panel (Thermo Fisher Scientific, Waltham, MA, USA) with two primer pools was designed following the manufacturer’s protocol to cover the entire human enterovirus genome. The HEV Ion AmpliSeq Custom Panel was designed with 567 amplicons, split into Pool 1 (284 amplicons) and Pool 2 (283 amplicons). The amplicon range was 125 to 375 bp, providing 99.94% coverage according to the reference genome.

cDNA synthesis, library preparation, templating, and sequencing were executed using the HEV Ion AmpliSeq Custom Panel and the Ion Torrent Genexus Integrated Sequencer, following the manufacturer’s instructions (Thermo Fisher Scientific, Waltham, MA, USA). Sixteen HEV samples were sequenced per lane using the Ion Torrent GX5 chip. Sequencing reads were processed, and their quality was assessed using Genexus software version 6.6.2 Revision F.0 (Publication Number MAN0017910, Thermo Fisher Scientific, Waltham, MA, USA). A custom assay definition file was employed for initial QC, covering chip loading density, median read length, and the number of mapped reads. Notably, the Genexus system automates all QC and data analysis steps post-sequencing, including mapping, variant calling, and optional report generation in a single day. The sequencing reads were mapped on the reference genome of HEV using the Torrent Mapping Alignment Program (TMAP), and the variant calling was performed using the CLC Genome workbench (Qiagen, Hilden, Germany). The complete genomes were named CVA6/strain number/GJ/KOR/2022, and the genome sequences were deposited in NCBI GenBank (accession No PP191111-PP191126).

### 2.4. Phylogenetic Analyses

For the global strains, 67 VP1 sequences and 59 3D polymerase (3Dpol) sequences from Finland, France, Spain, Japan, and China were obtained from the NCBI GenBank database. Multiple genome sequences were aligned using the Cluster-W algorithm in MEGA X software (version 10.1.8). A phylogenetic tree was constructed by the maximum likelihood (ML) method and the best-fitting nucleotide substitution model for ML was tested in MEGA X. For the analysis, a Kimura 2-parameter model with a variation rate among sites given by gamma distributed with invariant sites (G + I) was selected as the nucleotide substitution model according to the lowest Bayesian information criterion scores. During phylogenetic tree construction, statistical support for tree nodes was assessed using 1000 bootstrap replicates.

### 2.5. Mutation Analyses

A reference CV-A6 genome (KM114057), 7423 nucleotides in length, was obtained from the NCBI GenBank and used to identify mutations in the genomes of Gwangju strains.

## 3. Results

According to the EV laboratory surveillance system in Gwangju in 2022, the number of manifestations suggestive of HFMD infection began to increase rapidly at week 32 and continued until week 38 (Figure 1). In total, 277 clinical specimens were collected from patients with HFMD. Of these, 148 were male (53.4%; 148/277) and 129 were female (46.6%; 129/277), with a male-to-female sex ratio of 1.15:1. Patient age ranged from three months to 15 years, although most patients were less than five years old. Of these, 251 (90.6%; 251/277) were under 5 years of age; 224 (80.9%; 224/277) were 1–4 years old and the other 27 (9.7%; 27/277) were less than one year old (Table 1). Among the 277 cases of HFMD with laboratory results, 93.5% (259/277) had fever. Pharyngitis and common cold, which align with respiratory symptoms, and vomiting, diarrhea and abdominal pain, which are considered the gastrointestinal symptoms, were also reported in 5.4% (15/277) and 1.4% (4/277) of patients, respectively (Table 1). Of the 277 specimens, 82.3% (228/277) were positive for HEV. The main genotype detected in the HEV-positive samples was CV-A6, which accounted for 95.6% cases (218/228), whereas the remaining 10 samples were not typed (4.4%; 10/228).

In total, 16 complete genomes of CV-A6 were obtained in this study. Recombination events were not observed. Phylogenetic analysis of CV-A6 was performed by aligning its genome sequences. The 16 CV-A6 strains in this study and another 67 global CV-A6 strains from GenBank were analyzed using the sequences of their VP1 capsid region. A recent study has shown that CV-A6 variants could be categorized into four major groups, namely, A–D genotypes, and that genotype D could be further divided into three sub-genotypes (D1–D3) using VP1 gene analysis [18]. Following the above classification, phylogenetic analysis revealed that CV-A6 strains from this study were all grouped under sub-genotype D3 (Figure 2). All Gwangju strains displayed close genetic relationships with the strains collected in China between 2020 and 2021. 

To elucidate mutations at the amino acid level of the VP1 gene, we compared 16 sequences from this study with that of the Finland strain (KM114057), which was reported as the sub-genotype D3 of CV-A6 [19]. Nineteen amino acid substitution positions were detected in the VP1 gene (A5T, S27N, T28I, V30A/D, T32S/N, S34Y, L35M, A45V, E47V/D, S97N/I, L98F, D99N, W104R, G110D, F111L, V112G, S137N, L215V, and T283A). All 16 strains harbored four mutations (A5T, S27N, S137N, and T283A).

Emerging new variants of CV-A6, which harbored 3Dpol regions that are phylogenetically distinct from other HEV-A types via recombination, may generate genetic diversity in the virus population. CV-A6 strains were assigned recombinant forms (RFs) based on the 3Dpol coding region [11,20,21]. Accordingly, all 16 CV-A6 strains prevalent in Gwangju clustered within the previously assigned recombinant RF-A group (Figure 3). RF-A was first described in Finland in 2008 and variants within the RF-A group were subsequently detected in Asia. Based on the 3Dpol sequences, the recombinant variant RF-A was determined to be the predominant form in Gwangju. Taken together, we found that sub-genotype D3 and RF-A was the predominant genotype of CV-A6 circulating in Gwangju during the late summer and early autumn of 2022.

## 4. Discussion

HFMD outbreaks and epidemics have become a critical public health threat to younger children. In this study, the major population susceptible to HFMD in Gwangju includes children under five years of age. This is similar to the results of previous epidemiological studies on HFMD [22,23]. Our analysis of weekly reported number of HFMD cases revealed that the HFMD incidence commonly showed seasonal peaks, with a large peak from late spring to early summer, followed by a smaller peak in autumn. In 2019, HFMD incidence showed a similar seasonal pattern with cases increasing from May and peaking in July. After the outbreak of COVID-19, the incidence of HFMD declined sharply and occurred rarely in South Korea in 2020 and 2021. However, in 2022, a resurgence of the HFMD epidemic was observed in Gwangju, and the seasonal characteristics changed. Compared to that observed in the previous year of 2019, the peak period shifted from May–June to August–October. The HFMD epidemic started in week 27 of 2022, whereas it started between weeks 19 and 20 of 2019, thereby revealing a delayed start of at least 8 weeks. In late summer and early fall of 2022, we identified a sharp increase in the number of HFMD cases in Gwangju. This was the first HFMD epidemic to occur since the COVID-19 pandemic.

Our laboratory surveillance revealed that this atypical epidemic was primarily caused by CV-A6 infection and no other types of EV could be identified. EV-A71 and CV-A6 caused a major epidemic in 2019 and not EV-A71 but CV-A6 was rarely detected in 2020 [24]. The degree of sequence similarity in VP1 region at both the nucleotide and amino acid levels was used as a criterion for the identification and assignment of distinct EV subtypes. CV-A6 strains have been assigned to four genotypes, A–D, and sub-classified into seven sub-genotypes (B1–B2, C1–C2 and D1–D3) based on VP1 sequences. CV-A6 prototype strain Gdula (AY421764) from the United States formed a single branch that was denoted as genotype A [1]. Genotypes B and C comprised few isolates from China and India, respectively. Sub-genotype D1 consisted of strains from Japan, Australia, Taiwan, Spain, and France. Sub-genotype D2 was primarily assigned to isolates from China, with the exception of one strain isolated from Japan. Sub-genotype D3 included strains from Finland, Taiwan, Spain, France, Germany, Australia, Japan, and China. In this study, molecular and phylogenetic analyses of VP1 sequences showed that all strains from Gwangju clearly belonged to the sub-genotype D3. Since 2008, the D3 strains have become predominant in many countries throughout the world [8,14,17,18,22]. Therefore, the transmission of the sub-genotype D3 may have been responsible for persistent global outbreaks of CV-A6-associated HFMD. Analysis of the VP1 gene not only distinguishes genotypes but also allows estimation of infectivity, antibody responsiveness, and other factors. However, in this study, genotyping failed for approximately 4.4% (10 out of 228) of the samples diagnosed as HEV positive. The reason for this failure is attributed to the lower viral load (ct 30~ct 33) in these samples, resulting in a difference in sensitivity between diagnostic primers and genotyping primers. Consequently, the sensitivity of the genotyping primers was insufficient for confirmation. These findings are consistent with those reported by Liu et al. [25]. 

In addition, CV-A6 strains from Gwangju had 19 amino acid alterations including A5T, V30A, S137N, and T283A in the VP1 gene compared to the Finland strain (KM114057). CV-A6 strains circulating in Guangxi from 2010 to 2017 also had six amino acid switches, namely, A5T, V30A, S137N, V174I, I242V, and T283A, which might be associated with the CV-A6 related HFMD pandemic [26]. However, further study is needed to validate the relationship of these amino acids changes and the infectivity of CV-A6.

Nevertheless, phylogenetic and molecular analyses only based on the VP1 sequences have some limitations because of recombination. Recombination events are frequently observed within EV genomes and commonly occur in non-structural protein-coding regions of EVs. Among the non-structural protein domains, the 3Dpol coding region encodes RNA-dependent RNA polymerase and is essential for virus replication, which implies that recombination of the 3Dpol region may lead genetic diversity of EVs. According to the phylogenetic analysis of 3D polymerase sequences, CV-A6 strains have been assigned into RF-A to X [14]. Although recombination events were absent in the genomes obtained, RF-A was the prevalent recombinant variant in this study, thereby suggesting that D3/RF-A was the predominant form in Gwangju. 

The detection of a single CV-A6 subclade may have been affected by implementation of public health measures to control the SARS-CoV-2 during the pandemic period, which may have led to mitigate the circulation of other EVs. Our surveillance previously reported that nine EV genotypes were identified for the HFMD cases in 2019 [24]. Among them, the most predominant type was EV-A71, followed by CV-A6, CV-A16, CV-A10, CV-A2, E9, CV-A5, E21, and CV-A9. During 2020–2021, extremely low levels of EVs were detected in South Korea owing to strict public health interventions, such as school/kindergarten closure, social distancing, mandatory use of face masks, and maintenance of hand hygiene. Therefore, non-pharmaceutical interventions over the past 2 years may have reduced EV transmission and resulted in a lack of immune stimulation. Reduction in the number of infected or immunized persons may lead to higher susceptibility to viral infections, including EV infections. A similar phenomenon has been observed in the reported outbreaks of HFMD in France and Brazil in 2021, which were larger than those before the COVID-19 pandemic and associated with a single lineage of CV-A6 [22,27].

This study highlights the re-emergence of CV-A6-associated HFMD after the relaxation of social mitigation. Therefore, CV-A6 infections should be monitored continuously to understand the increased risk during the COVID-19 pandemic. Moreover, the genomic characteristics of CV-A6 should be shared and analyzed to enable the interpretation of the alterations in its transmissibility, infectivity, and pathogenicity. Therefore, continued surveillance of circulating EV types is required to monitor the pathogen spectrum of HFMD and its epidemiological trend. Surveillance for HFMD may provide valuable information for the prevention and control of the disease. Further laboratory and clinical investigations are essential for managing HFMD. Public health interventions and control measures against COVID-19 have suppressed the occurrence of HFMD. Based on these findings, we concluded that the prevention of EV infections by following personal protective measures is important for reducing the impact of HFMD on health.

## Figures and Tables

**Figure 1 viruses-16-00476-f001:**
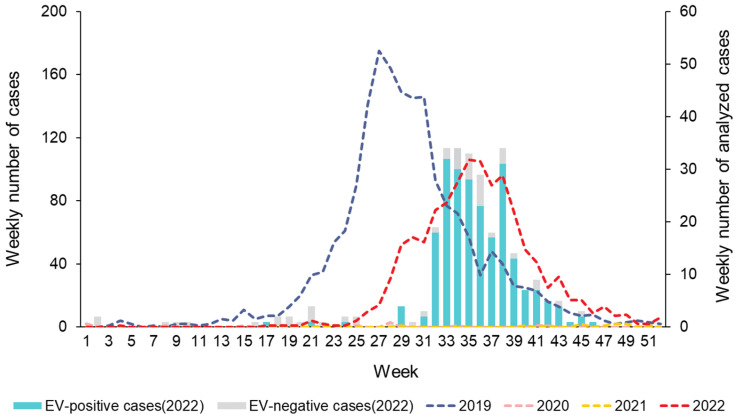
Weekly number of HFMD cases reported by the sentinel surveillance from the Korea Disease Control and Prevention Agency (KDCA), 2019−2022 (n = 3479), and collected clinical samples with HFMD from the enterovirus laboratory surveillance system in Gwangju, 2022 (n = 277).

**Figure 2 viruses-16-00476-f002:**
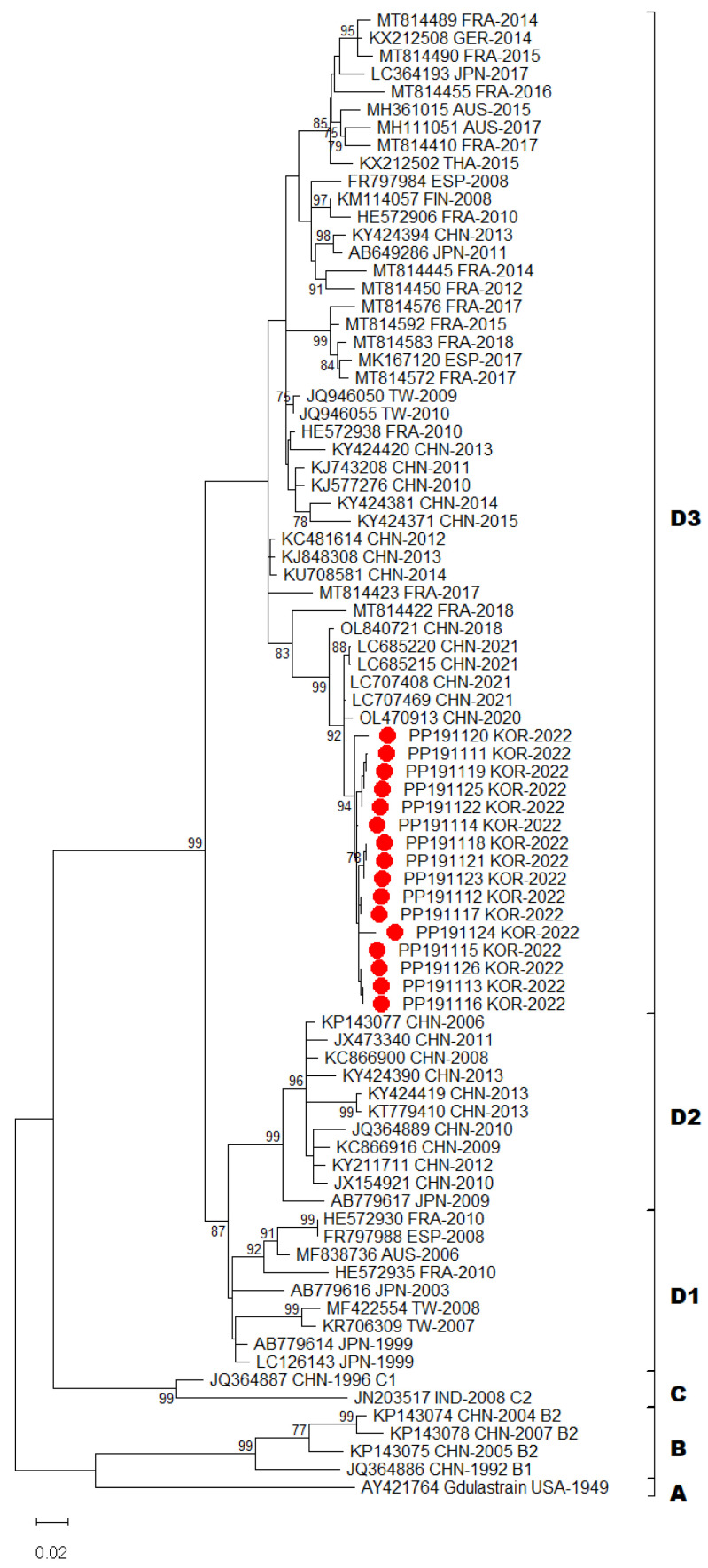
Phylogenetic tree of partial VP1 region with CV-A6 sequences (657 bp; CV-A6 Gdula strain VP1 nucleotide positions 2630–3286). Tree was constructed using the maximum likelihood method with the Kimura-2 parameter model of MEGA X. Bar denotes the evolutionary distance according to the number of nucleotide substitutions per site. Bootstrap analysis was performed with 1000 replicates. Bootstrap values lower than 70% are not shown. For clarity, the 16 CV-A6 sequences obtained in this study are highlighted with red dots.

**Figure 3 viruses-16-00476-f003:**
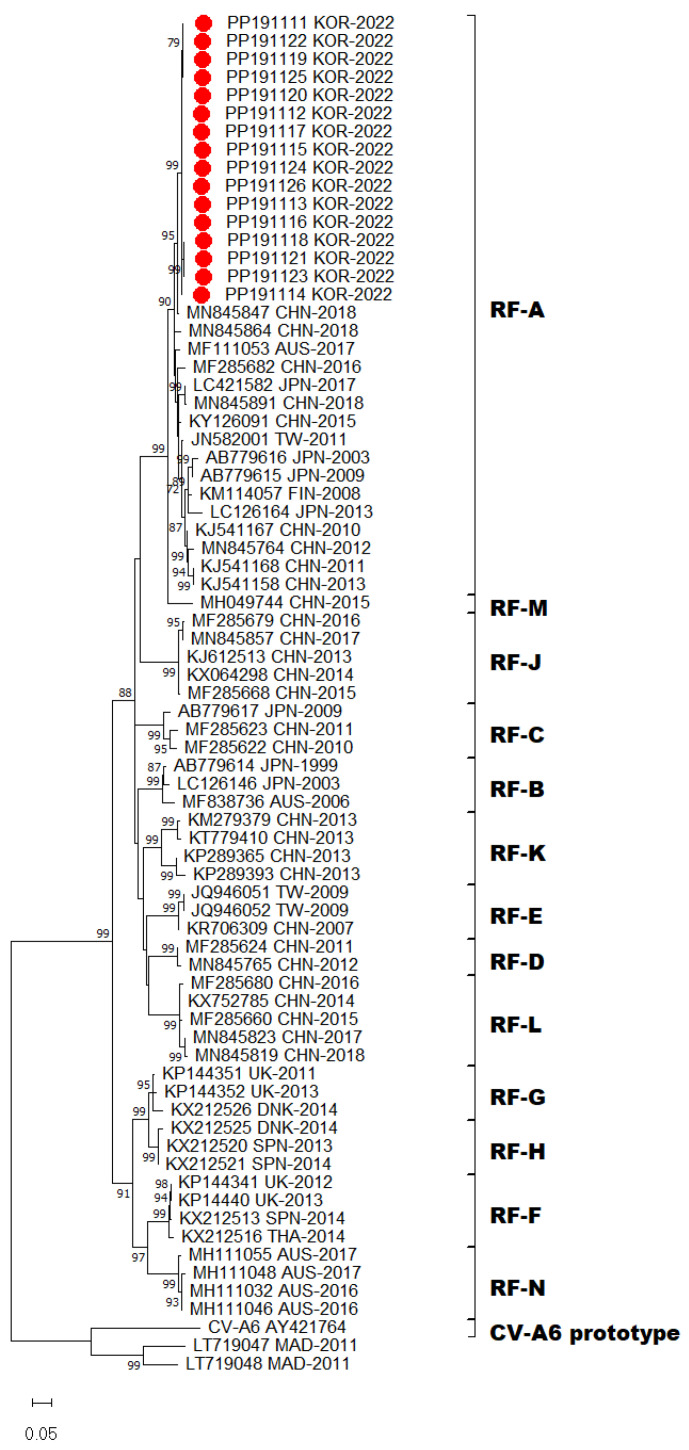
Phylogenetic tree of partial 3Dpol region with CV-A6 sequences (1161 bp; CV-A6 Gdula strain 3Dpol nucleotide positions 6188–7348). Tree was constructed using the maximum likelihood method with the Kimura-2 parameter model of MEGA X. Bar denotes the evolutionary distance according to the number of nucleotide substitutions per site. Bootstrap was performed with 1000 replicates. Bootstrap values lower than 70% are not shown. For clarity, the 16 CV-A6 sequences obtained in this study are highlighted with red dots.

**Table 1 viruses-16-00476-t001:** Demographic characteristics of hand, foot, and mouth disease cases, Gwangju, 2022.

		Total (n = 277)	%
Gender	Male	148	53.4
	Female	129	46.6
Age group (years)	<1	27	9.7
	1 to 4	224	80.9
	5 to 9	21	7.6
	>9	5	1.8
Symptoms	Fever	259	93.5
	Respiratory	15	5.4
	Gastrointestinal	4	1.4
	Unknown	10	4.4

## Data Availability

The data presented in this study are available on request from the corresponding author.

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
