# Peer review of "Epidemiological and Genetic Characterization of Coxsackievirus A6-Associated Hand, Foot, and Mouth Disease in Gwangju, South Korea, in 2022"

_viruses, 2024, doi:10.3390/v16030476_

Round 1
Reviewer 1 Report
Comments and Suggestions for Authors
This is an interesting report on the alteration of circulation of coxsackievirus A6 (CV-A6) and association with hand, foot, and mouth disease (HFMD) after the COVID-19 public health measures were relaxed in 2022 in part of South Korea. The authors report an alteration of the seasonal incidence of HFMD and that the typed enterovirus isolates were all CV-A6 belonging to the subgenotype and recombinant group circulating in China in 2020-1.
A minor issue is the statement on lines 186-187 “Our laboratory surveillance revealed that this atypical epidemic was primarily caused by CV-A6 infection and that other types of EV were not present.” In lines 126-128, the authors state: “The main genotype detected in the HEV-positive samples was CV- A6, which accounted for 95.6% cases (218/228), whereas the remaining 10 samples were not typed (4.4%; 10/228).” For this reason, the statement on lines 186-187 would be more accurate as: “Our laboratory surveillance revealed that this atypical epidemic was primarily caused by CV-A6 infection and no other types of EV could be identified.”
Author Response
- Reviewer’s comment: A minor issue is the statement on lines 186-187 “Our laboratory surveillance revealed that this atypical epidemic was primarily caused by CV-A6 infection and that other types of EV were not present.” In lines 126-128, the authors state: “The main genotype detected in the HEV-positive samples was CV- A6, which accounted for 95.6% cases (218/228), whereas the remaining 10 samples were not typed (4.4%; 10/228).” For this reason, the statement on lines 186-187 would be more accurate as: “Our laboratory surveillance revealed that this atypical epidemic was primarily caused by CV-A6 infection and no other types of EV could be identified.”
Author’s response: We appreciate the reviewer’s suggestion. We revised the manuscript following the reviewer’s comments.
(Page 8, lines 201-202)
Our laboratory surveillance revealed that this atypical epidemic was primarily caused by CV-A6 infection and no other types of EV could be identified.
Reviewer 2 Report
Comments and Suggestions for Authors
This manuscript describes the characterization of the HFMD outbreak in Korea in 2022. The authors identified CV-A6 as the main type of EV in the outbreak and characterized nearly complete genomes of some of the isolates.
Specific comments:
L73: Primer sequences used in the kits should be described. It is not clear how this kit could detect EVs.
Figure 1 and L181: There is no description of the 2019 outbreak. The authors should cite literature and explain the identities of the 2019 outbreak and possibly related CV-A6 in the 2022 outbreak.
L152 and 191: Literature describing the genotypes/clades of CV-A6 should be provided.
L156: A more detailed explanation of the recombinant forms of CV-A6 should be given.
L197: Literature should be added for the prevalence of the D3 strains.
L199: Are these D3 strains RF-A?
L204: Are these strains related to the 2022 outbreak? The relationship between the CV-A strains isolated in 2010-2017 and 2022 should be discussed in detail.
L217: This sentence is unclear and should be rephrased.
L224: This sentence is unclear in the context of this paragraph and needs to be rephrased. Are there any significantly different factors between Korea and France?
Author Response
- Reviewer’s comment: L73: Primer sequences used in the kits should be described. It is not clear how this kit could detect EVs.
Author’s response: We acknowledge the reviewer’s point of view on the inadequacy of information for the primer sequences we used. According to the reviewer’s comment, following sentences have been added to Materials and Methods section (page 2).
(page 2, lines 76-83)
Primers used in first PCR were as follows: forward primers 5'-GCR ATG TTR GGR ACW CAT GT-3'; 5'-GCS ATG TTR GGM ACR CAY GT-3' and reverse primer 5'-GGR TTB GWK GAN GTY TGC CA-3'. The primer combination used in the semi-nested PCR consists of two forward primers and two reverse primers. Primers used in secondary PCR were as follow: forward primers 5'-CCH GCD CTH ACC GCW GTG GAR ACD GG-3'; 5'-CCM ATM CTH CAA GCH GCH GAG GAG AYY GG-3' and reverse primers 5'-GGR SCN CCD GGW GGY ACA WAC AT-3'; 5'-GGH GCV CCY GGY GGY ACR TAC AT-3'.
- Reviewer’s comment: Figure 1 and L181: There is no description of the 2019 outbreak. The authors should cite literature and explain the identities of the 2019 outbreak and possibly related CV-A6 in the 2022 outbreak.
Author’s response: We appreciate the valuable comments from the reviewer. In accordance with the reviewer’s comment, the following sentences and reference have been added into Discussion section (page 8).
(Page 8, lines 191-193)
In 2019, HFMD incidence showed a similar seasonal pattern with cases increasing from May and peaking in July.
(Page 8, lines 203-204)
EV-A71 and CV-A6 caused a major epidemic in 2019 and not EV-A71 but CV-A6 rarely detected in 2020 [24].
(Reference)
[24] Kim, M.J.; Lee, J.-e.; Kim, K.G.; Park, D.W.; Cho, S.J.; Kim, T.S.; Kee, H.-y.; Kim, S.-H.; Park, H.j.; Seo, M.H. Long-term sentinel surveillance of enteroviruses in Gwangju, South Korea, 2011–2020. Scientific Reports 2023, 13, 2798.
- Reviewer’s comment: L152 and 191: Literature describing the genotypes/clades of CV-A6 should be provided.
Author’s response: We appreciate the reviewer’s comment. Based on the reviewer’s comment, we have added references describing the genotypes of CV-A6.
(Page 6, line 163)
To elucidate mutations at the amino acid level of the VP1 gene, we compared 16 sequences from this study with that of Finland strain (KM114057), which was reported as the sub-genotype D3 of CV-A6 [19].
(reference)
[19] Österback, R.; Koskinen, S.; Merilahti, P.; Pursiheimo, J.-P.; Blomqvist, S.; Roivainen, M.; Laiho, A.; Susi, P.; Waris, M. Genome sequence of coxsackievirus A6, isolated during a hand-foot-and-mouth disease outbreak in Finland in 2008. Genome announcements 2014, 2, 10.1128/genomea. 01004-01014.
(line 209)
CV-A6 prototype strain Gdula (AY421764) from the United States formed a single branch that was denoted as genotype A [1].
(Reference)
[1] Oberste, M.S.; Peñaranda, S.; Maher, K.; Pallansch, M.A. Complete genome sequences of all members of the species Human enterovirus A. Journal of General Virology 2004, 85, 1597-1607.
- Reviewer’s comment: L156: A more detailed explanation of the recombinant forms of CV-A6 should be given.
Author’s response: We acknowledge the reviewer’s point of view on the inadequacy of explanation of the recombinant forms of CV-A6. We revised our manuscript by providing additional descriptions into Results section (page 6).
(Page 6, lines 167-169)
Emerging new variants of CV-A6, which harbored 3Dpol regions that are phylogenetically distinct from other HEV-A types via recombination, may generate genetic diversity in the virus population. CV-A6 strains were assigned recombinant forms (RFs) based on the 3Dpol coding region [11,20,21].
- Reviewer’s comment: L197: Literature should be added for the prevalence of the D3 strains.
Author’s response: We appreciate the reviewer’s suggestion. We have added references for the prevalence of the D3 strains (page 8).
(Page 8, lines 216-217)
Since 2008, the D3 strains have become predominant in many countries throughout the world [8,14,17,18,22].
- Reviewer’s comment: L199: Are these D3 strains RF-A?
Author’s response: We appreciate the reviewer’s comment. Though RF-A of D3 stains have turned out to be the predominant type in many countries, such as Finland (2008), France (2010-2018), China (2017-2019) and Thailand (2014-2015 and 2019-2022), D3 stains have underwent recombination events, generating other recombinant forms. Many studies have reported that other recombinant variants had been prevalent in China (RF-J and -L; 2013-2016), Denmark (RF-G; 2003-2011), United Kingdom (RF-G; 2003-2011) and Thailand (RF-F; 2014, RF-Y; 2019 and RF-N; 2022). Therefore, the prevalent recombinant forms of D3 strains are varied depending on the country and year.
- Reviewer’s comment: L204: Are these strains related to the 2022 outbreak? The relationship between the CV-A strains isolated in 2010-2017 and 2022 should be discussed in detail.
Author’s response: We appreciate the valuable comments from the reviewer. In that paragraph, we attempted to describe the amino acid alterations of VP1 region observed in this study. Because the VP1 protein of EVs plays an essential role in evading the host’s immune response, analysis of amino acid exchanges in the VP1 coding region suggested that CV-A6 had changed its infectivity of the host and contributed to the development of a vaccine and anti-EV drugs targeting capsid. In past report, the structure of CV-A6 was analyzed, and it was reported that five VP1 loops (BC, DE, EF, GH and HI) seem to be excellent targets for designing specific anti-CV-A6 vaccines. Some substitution sites (A5T, V30A, S137N and T283A) observed in this study are also have been reported from other D3 representatives not only China (2010-2017) but also Vietnam (2011-2015). Among them, S137N substitution is located in the DE loop and will add N-glycosylation site, but its function still remained to be studied.
Though it is hard to explain the direct relationship between the CV-A6 strains isolated from Guangxi and Gwangju, sharing the same amino acid mutations in VP1 region may be associated with persistent international circulation of CV-A6. Therefore, further study is needed to validate the relationship of these amino acids changes and the infectivity of CV-A6. To clarify the point made by the reviewer, we added following sentences in Discussion section (page 8).
(Page 8, lines 224-225)
However, further study is needed to validate the relationship of these amino acids changes and the infectivity of CV-A6.
(reference) Xu, L., Zheng, Q., Li, S. et al. Atomic structures of Coxsackievirus A6 and its complex with a neutralizing antibody. Nat Commun 8, 505 (2017). https://doi.org/10.1038/s41467-017-00477-9
- Reviewer’s comment: L217: This sentence is unclear and should be rephrased.
Author’s response: We really appreciate the reviewer’s comment on improving the quality of the manuscript. We revised the manuscript following the reviewer’s comments and suggestions.
(Pages 8-9, lines 236-243)
The detection of a single CV-A6 subclade may have been affected by implementation of public health measures to control the SARS-CoV-2 during the pandemic period, which have led to mitigate the circulation of other EVs. Our surveillance previously reported that 9 EV genotypes were identified for the HFMD cases in 2019 [24]. Among them, the most predominant type was EV-A71, followed by CV-A6, CV-A16, CV-A10, CV-A2, E9, CV-A5, E21 and CV-A9. During 2020-2021, extremely low levels of EVs were detected in South Korea owing to strict public health interventions, such as school/kindergarten closure, social distancing, mandatory use of face masks, and maintenance of hand hygiene.
(Reference)
[24] Kim, M.J.; Lee, J.-e.; Kim, K.G.; Park, D.W.; Cho, S.J.; Kim, T.S.; Kee, H.-y.; Kim, S.-H.; Park, H.j.; Seo, M.H. Long-term sentinel surveillance of enteroviruses in Gwangju, South Korea, 2011–2020. Scientific Reports 2023, 13, 2798.
- Reviewer’s comment: L224: This sentence is unclear in the context of this paragraph and needs to be rephrased. Are there any significantly different factors between Korea and France?
Author’s response: We really appreciate the reviewer’s constructive criticisms aimed at improving the quality of the manuscript. In that paragraph, we attempted to explain the irregular occurrence of HFMD in 2022 and why the single subclade of CV-A6 was prevalent in Gwangju, by referring the similar cases that occurred in other countries. Similar phenomenon coincided with HFMD outbreaks reported from other countries during pandemic period, including France and Brazil in 2021. Magnitude of HFMD epidemic was larger than that before the COVID-19 pandemic (France) and single CV-A6 lineage was prevalent during the pandemic period (Brazil). To clarify the point made by the reviewer, we revised our manuscript (page 9).
(Page 9, lines 243-249)
Therefore, non-pharmaceutical interventions over the past 2 years may have reduced EV transmission and resulted in lack of immune stimulation. Reduction in the number of infected or immunized persons may lead to higher susceptibility to viral infections, including EV infections. Similar phenomenon has been observed by the reported outbreaks of HFMD in France and Brazil in 2021, which was larger than that before the COVID-19 pandemic and associated with single lineage of CV-A6, respectively [22,26].
Reviewer 3 Report
Comments and Suggestions for Authors
The authors examined samples from Korean children suffering from foot-and-mouth disease for enteroviruses. CV-A6 was found in the majority of 2022 cases. Complete CV-A6 sequences could be determined from sixteen samples, all of which could be assigned to the D3 clade on the basis of the VP1 sequence.
The work is of medical relevance, but some aspects are unclear.
1) What clinical materials were used for EV detection?
2) The description of symptoms in Table 1 is incomplete; in my opinion, pharyngeal symptoms are not respiratory symptoms, and it remains unclear what the authors mean by digestive symptoms.
3) What types of EV were found in the four samples that were EV-positive but not CV-A6 (see also Discussion, page 8)? What happened to the VP1 sequences that were generated by nested PCR? Apparently, all but four of the VP1 sequences were CV-A6 sequences. To what extent were contaminations, which can occur in nested procedures, excluded?
4) Only six complete sequences are mentioned in the introduction to chapter 2.2.
5) What was the criterion for using K2-G+I as a nucleotide substitution model and for using the ML algorithm for tree calculation?
6) The size of the sequenced region should be indicated in Figures 2 and 3.
7) What conclusions can be drawn from the amino acid exchanges observed in VP1 compared to the Finnish type strain? Do these also occur in other D3 representatives?
8) Have cultivation attempts been made to obtain isolates?
Author Response
- Reviewer’s comment: What clinical materials were used for EV detection?
Author’s response: We acknowledge the reviewer’s point of view on the inadequacy of information for clinical materials we used. We revised our manuscript by adding the specimen information into Materials and Methods section (page 2).
(Page 2, line 63)
All clinical specimens (including feces, throat swab and nasopharyngeal swab) from patients presenting with HFMD were collected by the hospitals in the Gwangju area and sent weekly to the Institute of Health and Environment Research for detecting HEV.
- Reviewer’s comment: The description of symptoms in Table 1 is incomplete; in my opinion, pharyngeal symptoms are not respiratory symptoms, and it remains unclear what the authors mean by digestive symptoms.
Author’s response: We appreciate the valuable comments from the reviewer. After reviewing the references, we have categorized symptoms of HFMD into two aspects: sore throat (pharyngitis) and common cold, which align with respiratory symptoms and vomiting, diarrhea and abdominal pain which are considered the gastrointestinal symptoms. Additionally, we have revised the term ‘digestive’ to ‘gastrointestinal’ to provide clarity.
(Page 3, lines 133-136)
Pharyngitis and common cold which align with respiratory symptoms and vomiting, diarrhea and abdominal pain which are considered the gastrointestinal symptoms, were also reported in 5.4% (15/277) and 1.4% (4/277) patients, respectively (Table 1).
- Reviewer’s comment: What types of EV were found in the four samples that were EV-positive but not CV-A6 (see also Discussion, page 8)?
Author’s response: We appreciate the comments from the Editor. We have added the following paragraph to the discussion section. Analysis of the VP1 gene not only distinguishes genotypes but also allows estimation of infectivity, antibody responsiveness, and other factors. However, in this study, genotyping failed for approximately 4.4% (10 out of 228) of the samples diagnosed as HEV positive. The reason for this failure is attributed to the lower viral load (ct 30~ ct33) in these samples, resulting in a difference in sensitivity between diagnostic primers and genotyping primers. Consequently, the sensitivity of the genotyping primers was insufficient for confirmation. These findings are consistent with those reported by Liu et al. [27] .
- Reviewer’s comment: What happened to the VP1 sequences that were generated by nested PCR?
Author’s response: We appreciate the comments from the reviewer. In this study, the VP3 region and the VP1 region, which have the smallest difference between the enteroviruses and show the most similar nucleotide sequences among the gene regions encoding the capsid protein of the enterovirus, were used in the first amplification range, and part of the portion amplified by the first PCR (first PCR product size: 815 bp). Then, only VP1 site was used as the second amplification range (nested PCR product size: 371 bp). Of course, there is a difference between each enterovirus in the sequence of the part to be amplified.
- Reviewer’s comment: Apparently, all but four of the VP1 sequences were CV-A6 sequences. To what extent were contaminations, which can occur in nested procedures, excluded?
Author’s response: We appreciate the valuable comments from the reviewer. To avoid contamination, we applied several precautions: use of at least two negative controls to exclude contamination; usage of filter tips for each sample; each phase of the experimental procedure (e.g., extraction of viral nucleic acids) was performed in different rooms; and to guarantee the accuracy of amplification, positive controls were included in each reaction.
- Reviewer’s comment: Only six complete sequences are mentioned in the introduction to chapter 2.2.
Author’s response: We appreciate the reviewer’s comment and apologize for this typographical error. We corrected the typo accordingly.
(Page 2, line 88)
Sixteen HEV samples underwent whole genome sequencing.
- Reviewer’s comment: What was the criterion for using K2-G+I as a nucleotide substitution model and for using the ML algorithm for tree calculation?
Author’s response: We appreciate the valuable comments from the reviewer. To find best-fit substitution model for our sequence data set, model selection procedure was conducted in Mega X. As the models with the lowest BIC scores (Bayesian Information Criterion) are considered to describe the substitution pattern the best in Mega X, K2+G+I was selected as the substitution model for VP1 and 3Dpol analysis. To clarify the point made by the reviewer, we revised our manuscript by providing additional descriptions into Materials and Methods section (page 3).
(Page 3, lines 113-117)
A phylogenetic tree was constructed by the maximum likelihood (ML) method and best-fitting nucleotide substitution model for ML were tested in MEGA X. For the analysis, Kimura 2-parameter model with a variation rate among sites given by gamma distributed with invariant sites (G+I) was selected as the nucleotide substitution model according to the lowest Bayesian information criterion scores.
- Reviewer’s comment: The size of the sequenced region should be indicated in Figures 2 and 3.
Author’s response: We appreciate the reviewer’s suggestion. Though we have obtained complete sequences of VP1 and 3Dpol (915 bp and 1,386 bp, respectively) from our sixteen samples, we have used partial sequences of them to analyze the same size of the sequences of global CV-A6 strains obtained from GenBank. In accordance with the reviewer’s comment, we have added the sequenced regions in Figure 2 and 3.
(lines 155-156, Figure 2)
Phylogenetic tree of partial VP1 region with CV-A6 sequences (657 bp; CV-A6 Gdula strain VP1 nucleotide positions 2,630-3,286).
(lines 178-179, Figure 3)
Phylogenetic tree of partial 3Dpol region with CV-A6 sequences (1,161 bp; CV-A6 Gdula strain 3Dpol nucleotide positions 6,188-7,348).
- Reviewer’s comment: What conclusions can be drawn from the amino acid exchanges observed in VP1 compared to the Finnish type strain? Do these also occur in other D3 representatives?
Author’s response: We appreciate the valuable comments from the reviewer. Because the VP1 protein of EVs plays an essential role in evading the host’s immune response, resulting in the main target of neutralizing antibody, analysis of amino acid exchanges in the VP1 coding region suggested that CV-A6 had changed its infectivity of the host and contributed to the development of a vaccine and anti-EV drugs targeting capsid. In past report, the structure of CV-A6 was analyzed, and it was reported that four VP1 loops (BC, DE, EF and HI) seem to be excellent targets for designing specific antiviral against CV-A6 (ref). Mutations in the VP1 region can change the structural characteristics of the capsid and alter the ability to bind to host receptors.
Some substitution sites (A5T, V30A, S137N and T283A) observed in this study are also have been reported from other D3 representatives in China (2010-2019) and Vietnam (2011-2015). Among them, S137N substitution is located in the DE loop and will add a new N-glycosylation site, but its function still remained to be studied. Therefore, further structural and functional studies would have profound implications for the development and implementation of antiviral against CV-A6.
(reference) Xu, L., Zheng, Q., Li, S. et al. Atomic structures of Coxsackievirus A6 and its complex with a neutralizing antibody. Nat Commun 8, 505 (2017). https://doi.org/10.1038/s41467-017-00477-9
- Reviewer’s comment: Have cultivation attempts been made to obtain isolates?
Author’s response: We appreciate the comments from the reviewer. We have routinely conducted enterovirus surveillance per every week. In the routine diagnostic system, enteroviruses are identified by PCR-based methods, combined with partial sequencing for genotyping. Cultivation has been considered to be the gold standard for virus detection, but this method has limited applications in fast diagnosis and genotyping of EV. Therefore, we have tried whole genome sequencing (WGS) of enteroviruses directly from clinical specimens. To this end, HEV-positive samples with the lowest Ct values (Ct under 20) were selected for WGS. (Positive cut-off: Ct-value <= 33.4). Without propagation in cell culture, we have successfully obtained sixteen whole genome sequences of CV-A6. We showed WGS can be used to correctly identify enterovirus genotypes from patient specimens with high viral load, as indicated Ct-values.
Round 2
Reviewer 3 Report
Comments and Suggestions for Authors
The authors have significantly revised the manuscript. My comments have all been taken into account.